A fish can change its stripes: investigating the role of body colour and pattern in the bluelined goatfish

Tosetto Louise louise.tosetto@mq.edu.au
Hart Nathan S.
Williamson Jane E.
School of Natural Sciences, Macquarie University, Wallumattagul Campus , North Ryde , NSW , Australia
Waiho Khor
Electronic publication date: 2024 Jan 29
Publication date: 2024
Volume: 12
Electronic Location ID: e16645
Received 2023 Aug 7; Accepted 2023 Nov 20
Copyright: ©2024 Tosetto et al.
Copyright year: 2024
Copyright holder: Tosetto et al.
License: This is an open access article distributed under the terms of the Creative Commons Attribution License, which permits unrestricted use, distribution, reproduction and adaptation in any medium and for any purpose provided that it is properly attributed. For attribution, the original author(s), title, publication source (PeerJ) and either DOI or URL of the article must be cited.
License URL: https://creativecommons.org/licenses/by/4.0/

Keywords: Chromatic signal, Achromatic signal, Communication, Mullidae, Physiological colour change

Funding: School of Natural Sciences (formerly the Department of Biological Sciences) at Macquarie University Holsworth Wildlife Research Endowment fund This work was supported by the School of Natural Sciences (formerly the Department of Biological Sciences) at Macquarie University and the Holsworth Wildlife Research Endowment fund. The funders had no role in study design, data collection and analysis, decision to publish, or preparation of the manuscript.

==============================
Bluelined goatfish (Upeneichthys lineatus) rapidly change their body colour from a white horizontally banded pattern to a seemingly more conspicuous vertically banded red pattern, often when foraging. Given the apparent conspicuousness of the pattern to a range of observers, it seems unlikely that this colour change is used for camouflage and instead may be used for communication/signalling. Goatfish often drive multispecies associations, and it is possible that goatfish use this colour change as a foraging success signal to facilitate cooperation, increase food acquisition, and reduce predation risk through a ‘safety in numbers’ strategy. Using a novel approach, we deployed 3D model goatfish in different colour morphs—white without bands, white with black vertical bands, and white with red vertical bands—to determine whether the red colouration is an important component of the signal or if it is only the vertical banding pattern, regardless of colour, that fish respond to as an indicator of foraging success. Use of remote underwater video allowed us to obtain information without the influence of human observers on the communities and behaviours of other fish in response to these different colours exhibited by goatfish. We found that conspecifics were more abundant around the black- and red-banded model fish when compared with the white models. Conspecifics were also more likely to forage around the models than to pass or show attraction, but this was unaffected by model colour. No difference in the abundance and behaviour of associated heterospecifics around the different models was observed, perhaps due to the static nature of the models. Some species did, however, spend more time around the red- and black-banded fish, which suggests the change in colour may indicate benefits in addition to food resources. Overall, the results suggest that the body colour/pattern of U. lineatus is likely a signalling tool but further work is required to explore the benefits to both conspecifics and heterospecifics and to further determine the behavioural functions of rapid colour change in U. lineatus.

Introduction

Animal colouration has several important ecological functions and many animals have the capacity to alter their individual colouration and pattern depending on social or environmental context (Aspengren, Sköld & Wallin, 2009). Morphological colour change typically occurs over long timeframes such as days or months (Sugimoto, 2002). By contrast, physiological, or rapid, colour change tends to happen more quickly on the order of milliseconds to hours (Duarte, Flores & Stevens, 2017). Rapid colour change provides individual plasticity with the potential for quick alterations and flexibility in colour and pattern depending on their behavioural requirements (Nilsson Sköld, Aspengren & Wallin, 2013). There are numerous reports of marine animals rapidly changing colour, with cephalopods considered the masters of this phenomenon. Cephalopods use dynamic colouration for camouflage (Hanlon, 2007) and as an intraspecific communication tool (Brown, Garwood & Williamson, 2012; Mäthger et al., 2009). Many studies on colour change in fishes have demonstrated a role in camouflage (Akkaynak et al., 2017; Allen et al., 2015; Ramachandran et al., 1996; Stevens, Lown & Denton, 2014; Watson, Siemann & Hanlon, 2014) but have also indicated that rapid alterations in fish colouration are used as conspicuous signals in courtship (Erisman & Allen, 2005; Svensson et al., 2006), competition (Berglund & Rosenqvist, 2001) and mimicry (Cheney et al., 2009). Some marine fishes use colouration in signalling, such as male minnows (Phoxinus phoxinus) where colourful ornaments may signal status, courting activity and superior quality (Kekäläinen et al., 2010). Moreover, some fishes adopt or enhance vertical banding patterns to communicate information (Berglund & Rosenqvist, 2001; Erisman & Allen, 2005; Galván-Villa & Hastings, 2018; Miller & Allen, 2006; Morris, Mussel & Ryan, 1995).

The bluelined goatfish (Upeneichthys lineatus) occurs along the southeastern coast of New South Wales and has the capacity for rapid colour change. U. lineatus can rapidly modify its colouration and body pattern from plain buff/white to a prominent dark red and white vertically banded pattern in less than ten seconds (Tosetto, Hart & Williamson, 2023). There are anecdotal reports of U. lineatus adopting a red colour when resting at night (personal observation) but during the day they are significantly more likely to shift colour and adopt a dark red pattern when actively feeding (Tosetto, Hart & Williamson, 2023), seemingly making them more conspicuous (Fig. 1). U. lineatus feed in a similar fashion to other goatfish using a shovelling action where their snout is buried in the substrate with their body angled more than 30 degrees relative to the substrate (Krajewski et al., 2006). This body position potentially restricts both their field of view and mobility, which may reduce their capacity to detect and escape from predators. When U. lineatus feed and adopt the red banded colouration, they regularly have conspecifics and associated fish following them, and a previous study found that heterospecifics will spend significantly longer interacting with U. lineatus that are displaying the dark red banding while foraging (Tosetto, Hart & Williamson, 2023).

Figure 1 The two distinct colour patterns displayed by Upeneichthys lineatus.

This is the same individual, photographed approximately 30 s apart.

Goatfish are zoobenthivores (benthic carnivores) and found throughout coastal areas in both temperate and tropical systems (Uiblein, 2007). Their capacity to form feeding aggregations through nuclear-follower foraging associations is well reported (Hanaway-Moore & Keeler, 2012; Lukoschek & Mccormick, 2002; Sazima et al., 2007). Group foraging increases the feeding success of follower fish by providing access to otherwise unobtainable prey (Ternes et al., 2018) while the nuclear fish (goatfish) may benefit through anti-predator mechanisms such as increased vigilance and dilution effects by the followers (Cresswell et al., 2003). Many studies, particularly in the case of goatfish, suggest that nuclear fish are identified by followers via bottom disturbance and the resulting sediment plume (Krajewski, 2009; Sazima et al., 2007). However, foraging associations may also be influenced by local enhancement, a process whereby individuals use social cues such as fish presence or body position to indicate the location of suitable food patches (Brown & Laland, 2003; Ryer & Olla, 1992; Waite, 1981). There is also evidence of fish using referential gestures (Vail, Manica & Bshary, 2013) to signal food and instigate collaboration but whether distinct changes in colour and pattern play a role in signalling foraging success has not been widely explored. It is possible that the dark red banding pattern exhibited by U. lineatus when foraging is an intra- or interspecific signal to indicate increased food acquisition via a ‘safety in numbers’ approach.

If the colour and pattern change exhibited by U. lineatus is indeed a signal, then potential receivers must have the sensory systems to detect this signal. If fish are using chromatic cues and it is the red colour providing the signal, then fish must possess visual pigments with the spectral sensitivity to discriminate those longer wavelengths (Endler, 1992; Marshall & Vorobyev, 2003). If it is an achromatic cue and receivers are using the contrast of the vertical banding as the signal, regardless of the colour, then fish must have the visual acuity to resolve those bands from an ecologically relevant distance (Cronin et al., 2014). Recent findings suggest U. lineatus are trichromats, with three cone pigments having wavelengths of maximum absorbance at 413, 494 and 524 nm, and can discriminate the ‘red’ colours when longer wavelengths of light are available (Tosetto et al., 2021). The visual acuity of U. lineatus is estimated at 6.2 cpd and the maximum distance from which the banding could be resolved in clear waters is approximately seven meters (Tosetto et al., 2021). Whether associated heterospecifics possess the spectral sensitivity to perceive longer (red) wavelengths or have the acuity to resolve the banding from a suitable distance is unknown. We do know that many coastal fish have visual pigments that are tuned to match the wavelengths of light that are most abundant in these habitats, typically in the short (blue) and medium (green) wavelength portion of the visible spectrum, and generally lack pigments that absorb strongly at longer (red) wavelengths of light (Bowmaker, 1984; Lythgoe & Partridge, 1991). Nevertheless, several coastal teleosts possess visual pigments that are similar to U. lineatus, so discrimination of red colouration is possible. The visual acuity of fish found in temperate coastal environments have not been widely published but fish inhabiting similar environments have visual acuity in the region of 10 cpd (Caves, Sutton & Johnsen, 2017), suggesting that the banding of U. lineatus may be resolved from 10 m away in suitable conditions. Given that the red colour cannot always be perceived, such as in low light when cone photoreceptors are not active or deeper water environments where there is little red light, and that some of the associated fish may not have the spectral sensitivity to discriminate the longer wavelengths of light, it is also possible that it is the change in patterning rather than the colour of U. lineatus that provides a meaningful signal.

In this study, we used three-dimensional (3D) printed model fish to investigate if changes in colour and / or patterning of U. lineatus play a role in signalling potential food patch locations to both conspecifics and associated heterospecifics. Such 3D printed models provide a way to produce identical objects while only manipulating the traits of interest (in this case colour) and preserving the ecological setting (Behm et al., 2018; Bulté et al., 2018). A similar approach has been employed across a range of behavioural studies, including using model bird eggs in evaluating egg rejection behaviour (Igic et al., 2015) and decoy female turtles to assess the effect of body size on mate choice (Bulté et al., 2018). Further, zebrafish have demonstrated similar attraction to both live conspecifics and 3D printed replicas (Ruberto, Polverino & Porfiri, 2017), suggesting that 3D models are an effective tool for assessing fish response to different colour morphs.

Here, we deployed 3D model goatfish in three different colour morphs; a red and white vertically banded model, a black and white vertically banded model and a plain white model with no banding to examine differences in fish behaviour and interactions in proximity to the different model variants. Using both a red striped and a black striped model fish allowed us to determine whether fish were responding to a chromatic or achromatic signal to discern between the colouration or the vertical banding. We also included a plain white model fish to control for interest in the 3D model fish. This study allowed us to remotely obtain information on the communities and behaviours of both conspecifics and heterospecifics in response to these different colour patterns via the use of remote underwater video, without the influence of human observers. Specifically, we examined whether U. lineatus or associated heterospecifics displayed differences in their behaviour (passing, attracted or foraging) in proximity to different coloured models. We also explored if the total amount of time that U. lineatus and associated heterospecifics spent around the different model fish varied. We predicted that both U. lineatus and associated heterospecifics would spend a greater amount of time around the red and black models when compared with the white models. We also predicted that if the red patterning is a foraging signal to conspecifics, then there would be a greater number of U. lineatus feeding near the red models than the black or the white models. Furthermore, because some coastal fishes may not be able to discriminate long wavelength reflecting colours, we expected that some associated heterospecifics would not be influenced by the colour and so anticipated no difference between the black and white banded model fish and the red and white banded model fish.

Materials & Methods

Ethics statement

This study was conducted in accordance with the Australian code for the care and use of animals for scientific purposes. All work with animals and methods for remote video recording were approved by Macquarie University Animal Ethics Committee (ARA 2016/020). Underwater video cameras were placed in isolated areas which were only accessible via boat thus, there were no human images captured on any of the videos obtained for this study.

3D model fish

Goatfish that had been euthanised for a previous experiment (following methods approved by the Macquarie University Animal Ethics Committee; ARA 2016/020) were photographed from the top, side and front and fine measurements taken so that the scale for a 3D model could be obtained. The 3D fish models (230 × 70 × 45 mm) were created using Cinema 4D (MAXON Computer Ltd.) and a Wavefront.obj file was generated for 3D printing (see Video S1). The model goatfish were then 3D printed using Polylactic Acid (PLA), a biodegradable thermoplastic. Models were sanded and sprayed uniformly with white plastic primer (Duramax Plastic Primer Spraypaint, Dulux 325 g) to remove any inconsistencies in surface texture from the printing process. Given that different individuals of U. lineatus exhibit essentially the same pattern when they change colour (Tosetto et al., 2021), we made one stencil that replicated the banded pattern for the model fish. The stencils were placed onto the six treatment model fish and sprayed with either black or red paint (three black and three red). Models were finished with a clear acrylic gloss (Dulux Duramax High Performance Enamel Gloss Clear) that was spray painted over their entirety. The three white control fish were also finished with the gloss finisher for consistency. Once completed, fish were soaked in running seawater for several days to ensure that any soluble chemicals emitting form the models were removed prior to use. Spectral reflectance measurements of the red and white, and the black and white bands of the 3D model fish were obtained using the spectrometer function of an underwater chlorophyll fluorometer (DIVING-PAM-II, Heinz Walz GnbH, Germany) (Fig. S1). The reflectance measurements for U. lineatus were obtained in a previous study (Tosetto et al., 2021).

Remote underwater video apparatus

The Remote Underwater Video (RUV) apparatus consisted of a T-shaped structure constructed with PVC (10 mm diameter). The structure was weighted by placing 12 mm reinforcing bar (Whites 12 × 600 mm rib reinforcing bar) inside the PVC piping and sealing each end with PVC caps. This expedited deployment into sediment and negated the need for additional weights on the RUV. A single GoPro™ digital video camera (Hero 2018 model) was placed 600 mm away from the model fish (Fig. 2). We did not use bait to attract fish as the aim of the study was to ascertain if there was a spontaneous effect of the different colour morphs on fish community structure and behaviours.

Figure 2 The 3D model fish deployed at various locations.

(A) There are two dolphins passing by the red and white model fish (B) An octopus passing the black and white model fish. (C) A leather jacket showing attraction to the white model fish (D) Schematic (not to scale) of the recording underwater video (RUV) apparatus with the GoPro™ indicated by the grey box.

Field sites

Experiments were conducted between July and September in 2018 and 2019 at 15 sites in New South Wales, Australia, from Sydney (35°52′39.77″S, 151°14′30.03″E) down to Eden (37°03′56.73″S, 149°53′59.09″E). Sites were chosen based on the presence of rocky reef with adjacent sand flats and the presence of U. lineatus. The RUVs were randomly deployed at five sites in Sydney, two sites in Botany Bay, six sites in Jervis Bay, one site in Kioloa, and one site in Eden (Fig. 3) (see Table S1 for details of all sites and GPS coordinates). Field work permits were issued by the Marine Fieldwork Manager at Macquarie University (TR-18-2893, TR-19-4335 and TR-19-4181).

Figure 3 Field sites across New South Wales.

Map showing locations of the 15 field sites in NSW. Insets show the five sites in Sydney and six sites in Jervis Bay.

Experimental design

Three locations positioned approximately 50 metres apart were chosen at each site. Because long wavelength (‘red’) light is attenuated strongly by water and therefore less abundant at depth, all test sites were in water between 5 and 10 metres deep, however, at each site the depth was the same for the three locations. The RUVs were deployed from a boat during relatively calm conditions so that suitable patches of sand adjacent to rocky reefs could be identified from the boat. At each of the three locations we placed a red-banded, a black-banded and a white model fish. Once each group of 3D fish was deployed, scuba divers then ensured correct positioning around the rocky reef and the burial of the PVC piping for each RUV, ensuring that no other part of the apparatus (apart from the GoPro camera) was visible (Fig. 2). Each RUV was placed so that the GoPro camera was near the reef looking out towards the 3D fish so that the fish was approximately 600–700 mm away from the reef. Each of the colour morphs was placed approximately five metres apart on different sides of the rocky reef so that models were not adjacent to each other. They were then buried into the sediment, thus making the fish snout rest on the sediment and obscuring the frame. The models were deployed for at least one hour with the GoPro recording in wide angle mode and at 30 frames per second. The RUVs were deployed once per day in the mornings (from 8 am–11 am).

Video analysis

Video footage was assessed using two different methods: (1) abundance and behaviour of individual conspecific or heterospecific fishes around the different colour models; and (2) the total time spent by conspecifics and associated heterospecific fish around each colour morph (time-in-view; TiV). Nine fish were identified as key heterospecifics; those species that were previously identified (Tosetto, Hart & Williamson, 2023) and those that had interacted most with our model fish in this study. Heterospecifics were identified using a field guide to fishes in southern Australia (Hutchins & Swainston, 1986). Key heterospecifics were; Ophthalmolepsis lineolatus (southern Maori wrasse), Acanthopagrus australis (yellowfin bream), Gerres subfasciatus (silver biddy), Chrysophrys auraus (juvenile snapper), Pseudocaranx georgianus (silver trevally), Eupetrichthys angustipes (snakeskin wrasse), Scobinichthys granulatus (rough leatherjacket), Atypicthys strigatus (mado) and Parupeneus spilurus (blacksaddle goatfish). While a highly abundant fish in this study was Trachurus novaezelandiae (yellowtail scad), they are a schooling fish and considered pelagic planktivores (Kingsford, 1989) and thus not a fish of interest in regards to U. lineatus. Trachurus novaezelandiae were included in total numbers of heterospecifics but were not identified as a key heterospecific for individual analysis.

Abundance and behaviour

Fish abundance was the total number of individual conspecifics and heterospecifics that interacted with the 3D model fish. For each interacting fish we recorded the species and its behaviour: passing, attracted or foraging. Passing was defined as no obvious interaction or interest in the model fish. Attracted was logged when an individual demonstrated some curiosity in the 3D model fish, such as stopping to look at the model fish, an obvious slowing in swim speed to investigate the model, or an obvious change in swimming direction towards the model fish. Foraging was recorded as any fish that probed or ate from the sediment within a 2 m radius of the model fish (Fig. 4).

Figure 4 Schematic of the fish behaviours recorded.

The different fish behaviours (passing, attracted and foraging) that were recorded for all individual conspecifics and heterospecifics in response to the 3D colour morphs. Each behaviour is shown as two examples in the schema.

We only recorded individuals where we could clearly see their eyes and markings and only one behaviour was recorded per individual. The scoring of behaviours was hierarchical whereby if an individual was both attracted to and fed near the model we recorded the behaviour as feeding given that the overall hypothesis is whether the change in colour and pattern is a foraging signal. Similarly, if it passed by and then demonstrated attracted behaviour, the attracted behaviour was recorded. If a fish left the field of view of the camera and a fish identical in size with distinct markings returned within 10 s, it was deemed to be the same fish and was not recorded again. If a similar looking fish returned into the screen after more than 10 s then it was recorded as a new fish unless clear markings identified it as the previous fish.

Time-in-view

Abundance and behaviour does not account for the time that a fish may have spent around a particular fish and only counts interactions at a single point in time. An assumption is that the dark red is a foraging signal to encourage group feeding associations and increase foraging success. Other potential benefits, such as increased vigilance, are harder to quantify, thus measuring the time that fish interact with the models may provide additional insights. To establish how much time a species spent around each colour morph, we used time-in-view. Time in view ((TiV) was recorded as the total time in seconds that at least one of a given species was in the frame (see Smith et al., 2011) for a detailed explanation of TiV). The TiV was recorded for U. lineatus and nine key heterospecifics. Fish were only counted as present in the frame when they were clearly in view. If a fish swam to the back of the frame and we could no longer see its eyes or markings, then it was deemed to be out of view. The frame that the fish entered was recorded and then the frame number that the fish left was recorded. We obtained the total time fish interacted with 3D models by counting total frames and dividing by the number of frames per second (30) to obtain the total seconds that the fish was in view.

Videos were analysed for 55 min with a two-minute acclimation time from the start of recording. This acclimation time allowed for sediment to settle and diver presence to dissipate. The computer software package EventMeasure was used to record species, behaviour and TiV (SeaGIS Pty Ltd., Bacchus Marsh, Australia).

Statistical analysis

Conspecific and associated fish behaviour

To evaluate differences in fish behaviour in response to different coloured model fish we first fitted a general linear fixed effects (glmer) model with Poisson distribution using the lme4 package in R (Bates et al., 2015). We included fish abundance as the response variable with fish behaviour (passing, attracted or foraging) and the colour variant of the goatfish model as fixed factors with an interaction term. Site was included as a random effect. To check for overdispersion we used the overdisp function (Gelman & Hill, 2006). Because the Poisson model did not fit well and was overdispersed, we constructed a linear mixed-effects model using the lme4 package to assess differences in behaviour between the different coloured models. Fish abundance was included as the response variable with behaviour and the 3D model colour as fixed effects and included site as a random effect. Because we were primarily interested in whether behaviour differed around the different coloured models, we specified an interaction between behaviour and model colour. The data were log(x+1) transformed and models met assumptions of linearity and homogeneity of variances with normal distribution of residuals. Significance of the interactions were confirmed using log likelihood ratios. The interaction term was dropped if not significant and the main effects of behaviour and model colour were assessed. Significance of the main effects were obtained from an ANOVA type III table using the anova function in the lmertest package (Kuznetsova, Brockhoff & Christensen, 2017). Pairwise comparisons among main effects were obtained using the pairwise method and a p-value adjustment equivalent to the Tukey test in the emmeans package (Lenth, 2022). This approach was used to explore differences in abundance and behaviour around the different models for both conspecifics and all associated heterospecifics. We also constructed models for each of the nine key heterospecifics identified above. All statistical analysis were completed in R Studio Version 2022.07.2 (R Core Team, 2022).

Time-in-view

We assessed if the time spent by conspecifics and associated heterospecifics around different models was influenced by model colour by fitting a linear mixed-effects model. The models were constructed in R using the lme4 package (Bates et al., 2015) with time spent around models by conspecifics and heterospecifics included as the response variable in all models. The 3D model colour was included as a fixed effect and the site included as a random effect. The data were log transformed and models met assumptions of linearity and homogeneity of variances with normal distribution of residuals. Significance of the main effects were obtained with an ANOVA type III table using the anova function in the lmertest package (Kuznetsova, Brockhoff & Christensen, 2017). Pairwise comparisons among main effects were obtained using the pairwise method and a p-value adjustment equivalent to the Tukey test in the emmeans package (Lenth, 2022).

Results

In total there were 58 species of fish that interacted with the different model fish. The species recorded, including their frequency around each of the coloured models, is outlined in Table 1.

Table 1 Frequency of each fish species recorded at each of the coloured models.

The totals provided are the number of individual fish recorded around each of the colour models and their different behaviours. Frequencies are listed in descending order according to the total number observed.

Species	Common Name	Total	3D Model Colour	Fish Behaviour	
			Black	Red	White	Attracted	Feeding	Passing	
Trachurus novaezelandiae	Yellowtail Scad	1080	364	437	279	73	18	989	
Gerres subfasciatus	Silver Biddy	539	137	193	209	66	277	196	
Upeneichthys lineatus	Bluelined Goatfish	418	126	199	93	62	198	158	
Ophthalmolepsis lineolata	Southern Maori Wrass	356	112	119	125	112	46	198	
Chrysophrys auratus	Snapper (juvenille)	249	66	77	106	139	36	74	
Atypichthys strigatus	Mado	196	85	22	89	138	0	58	
Acanthopagrus australis	Yellowfin Bream	195	120	52	23	9	41	145	
Scorpis lineolata	Silver Sweep	95	36	29	30	32	0	63	
Notolabrus gymnogenis	Crimson-banded Wrasse	88	24	34	30	13	2	73	
Eupetrichthys angustipes	Snakeskin Wrase	85	14	43	28	22	3	60	
Helotes sexlineatus	Eastern Striped Grunter	81	79	1	1	0	16	66	
Pseudocaranx georgianus	Silver Trevally	82	27	29	26	5	0	77	
Parupeneus spilurus	Blacksaddle Goatfish	80	30	30	20	30	9	41	
Pictilabrus laticlavius	Senator Wrasse	80	24	41	15	3	0	77	
Parma microlepis	White-ear	65	30	21	14	28	7	30	
Olisthops cyanomelas	Herring Cale	63	33	14	16	0	0	63	
Dinolestes lewini	Longfin Pike	61	3	49	9	8	0	53	
Achoerodus viridis	Eastern Blue Grouper	58	14	16	28	3	2	53	
Heterodontus portusjacksoni	Port Jackson Shark	48	11	16	21	4	0	44	
Brachaluteres jacksonianus	Southern Pygmy Leatherjacket	36	9	10	17	21	1	14	
Scobinichthys granulatus	Rough Leatherjacket	36	4	13	19	10	9	17	
Prionurus microlepidotus	Australian Sawtail	32	1	24	7	3	1	28	
Sillago ciliata	Sand Whiting	32	9	18	5	2	6	24	
Morwong fuscus	Red Morwong	29	6	15	8	11	6	12	
Neoodax balteatus	Little Weed Whiting	26	10	11	5	1	0	25	
Parapercis ramsayi	Spotted Grubfish	19	6	7	6	16	1	2	
Rhabdosargus sarba	Tarwhine	19	4	10	5	6	5	8	
Aplodactylus lophodon	Rock Cale	18	6	12	0	10	0	8	
Trygonoptera testacea	Common Stingaree	18	3	6	9	6	0	12	
Anoplocapros inermis	Eastern Smooth Boxfish	16	4	5	7	11	0	5	
Heteroscarus acroptilus	Rainbow Cale	16	6	7	3	3	0	13	
Meuschenia trachylepis	Yellowfin Leatherjacket	15	4	6	5	5	6	4	
Mecaenichthys immaculatus	Immaculate Damsel	14	0	0	14	5	3	6	
Myliobatis tenuicaudatus	Southern Eagle Ray	10	4	4	2	10	0	0	
Upeneichthys vlamingii	Bluespotted Goatfish	9	5	3	1	1	0	8	
Girella tricuspidata	Luderick	8	4	3	1	0	0	8	
Acanthaluteres spilomelanurus	Bridled Leatherjacket	6	0	5	1	6	0	0	
Bathytoshia brevicaudata	Smooth Stingray	6	1	3	2	4	0	2	
Haletta semifasciata	Blue Weed Whiting	6	1	1	4	1	0	5	
Pseudojuloides elongatus	Long Green Wrasse	5	0	2	3	0	0	5	
Trachinops taeniatus	Eastern Hulafish	5	0	5	0	1	0	4	
Meuschenia flavolineata	Yellowstriped Leatherjacket	4	1	2	1	0	3	1	
Trygonorrhina fasciata	Eastern Fiddler Ray	4	1	1	2	3	0	1	
Hypoplectrodes maccullochi	Halfbanded Seaperch	3	0	2	1	1	0	2	
Nemadactylus valenciennesi	Blue Morwong	3	2	1	0	1	0	2	
Urolophus cruciatus	Banded Stingaree	3	0	0	3	1	0	2	
Goniistius vestitus	Crested Morwong	3	1	1	1	0	0	3	
Aptychotrema rostrata	Eastern Shovelnose Ray	2	0	0	2	1	0	1	
Platycephalus fuscus	Dusky Flathead	2	0	1	1	0	0	2	
Aldrichetta forsteri	Yelloweye Mullet	1	0	1	0	0	0	1	
Cnidoglanis macrocephalus	Estuary Cobbler	1	1	0	0	0	0	1	
Dicotylichthys punctulatus	Threebar Porcupinefish	1	1	0	0	0	0	1	
Eubalichthys mosaicus	Mosaic Leatherjacket	1	0	1	0	0	0	1	
Latropiscis purpurissatus	Sergeant Baker	1	0	0	1	0	0	1	
Microcanthus strigatus	Stripey	1	0	0	1	0	0	1	
Notorynchus cepedianus	Broadnose Seven Gill Shark	1	0	0	1	1	0	0	
Scorpaena jacksoniensis	Eastern Red Scorpionfish	1	0	1	0	0	0	1	
Squatina albipunctata	Eastern Angelshark	1	0	1	0	0	0	1	

Abundance and behaviour—conspecifics (Upeneichthys lineatus)

There was no significant interaction between the behaviour of the goatfish and the three colour variants of the model fish (X2 = 3.400, P = 0.493). Examination of the main effects showed significant differences in behaviour (F = 19.885, P <0.001) with significantly more U. lineatus foraging around the model fish than those that showed attraction or passed the models, and significantly more U. lineatus passing the models than those that displayed attraction. There was also a significant difference between the model colours (F = 4.571, P = 0.011) with significantly more U. lineatus around the black and red models when compared with those around the white models. There was no difference in the number of U. lineatus between the black and red models (Fig. 5; see Table S2 for all pairwise comparisons).

Abundance and behaviour—associated heterospecifics

Overall, there was no significant interaction between behaviour of the heterospecifics and the colour of the model fish (X2 = 2.512, P = 0.642). Analysis of the main effects showed the colour of the model did not influence heterospecific abundance (F = 0.030, P = 0.970) but we did see overall differences in heterospecific behaviour (F = 32.943, P < 0.001) regardless of goatfish model colour. Associated heterospecifics fish passed the models significantly more than they were attracted (t = 2.776, P = 0.0160) or foraged (t = 8.065 P < .0001). There were also significantly more heterospecifics attracted to the models than foraged around the models (t = 5.085, P < 0.001) (Fig. 6).

Individual analysis of the nine key heterospecifics found no significant interactions between the model fish colour and their behaviour. Analysis of the main effects revealed some differences in behaviour regardless of model colour. O. lineolatus passed and showed attraction to the models significantly more times than they foraged around them (F = 5.370, P = 0.005), Atypicthys strigatus were attracted significantly more times than they foraged (F = 5.710, P = 0.004). Eupetrichthys angustipes passed the models significantly more times than fed near the models (F = 10.170, P < 0.001) and P. georgianus passed significantly more times than displayed attraction or fed near the models (F = 11.340, P < 0.001). All pairwise comparisons for the main effects are provided in Table S3.

Figure 5 Distribution of conspecific interactions with 3D models.

Violin plot showing U. lineatus interactions with each of the different coloured 3D model fish (the width of the shaded area represents the proportion of the data located there). Data points demonstrate interactions of individual fish. Differences in lowercase letters indicate significant differences (P < 0.05) in conspecific behaviour (n = 45).

Figure 6 Distribution of heterospecific interactions with 3D models.

Violin plot showing the distribution of all heterospecific interactions with each of the different coloured 3D model fish (the width of the shaded area represents the proportion of the data located there). Data points demonstrate interactions of individual fish. Differences in lowercase letters indicate significant differences (P < 0.05) in heterospecific behaviour (n = 45).

Time-in-view–all associated fishes (conspecifics and heterospecifics)

Overall, there was no difference in the time that conspecifics spent around the different models (F = 0.286, P = 0.751). There was also no difference in the time spent by all the heterospecifics around the models (F = 1.150, P = 0.317). However, some differences in the time spent around the different model fish were observed for juvenile C. auratus (F = 5.296, P = 0.006), S. granulatus (F = 6.141, P = 0.007), G. subfasciatus (F = 3.331, P = 0.038), O. lineolatus (F = 3.102, P = 0.046) and A. australis (F = 9.828, P = 0.006). Juvenile C. auratus and S. granulatus spent significantly longer around the red model fish than the black or white fish. Gerres subfasciatus spent longer around the black model fish than the white model fish and O. lineolatus and A. australis spent more time around the white model fish than the black model fish. There were no differences in time spent around the different model fish for A. strigatus (F = 3.411, P = 0.041), P. spilurus (F = 0.803, P = 0.461), E. angustipes (F = 0.917, P = 0.406) or P. georgianus (F = 1.699, P = 0.197) (Fig. 7. See Table S4 for all pairwise comparisons).

Figure 7 Comparison of the time that key heterospecifics spent around the different coloured 3D model fish.

Each box represents the interquartile range and median. Whiskers represent range of data within 1.5 times the interquartile range. Data points demonstrate time spent by individual fish (n = 45).

Discussion

In this study we assessed whether the striped body pattern of U. lineatus represents a foraging signal that encourages group feeding. We further sought to understand whether, if this is the case, the relevant signal perceived by the associated fish is achromatic or influence by colour. Overall, we observed 58 different fish species and over 4,000 individual fish interacting with the 3D models. There was no interaction between the behaviour of conspecifics and the model fish colour. We did observe significantly more foraging interactions by conspecifics around the model fish than passing or attracted interactions irrespective of colour. We also recorded more interactions with the red and black model fish when compared with the white models regardless of behaviour. However, no difference was observed when assessing the total time that conspecifics spent around the different models. Associated heterospecifics did not exhibit differences in behaviour in response to different models. We did record more passing interactions for some key heterospecifics when compared with foraging or attracted interactions around the model fish. There were also significant differences in the time that some fish species spent in the proximity of the different models, suggesting that the striped body pattern of U. lineatus may be providing an interspecific signal to some species.

The results suggest that conspecifics may be responding to the difference in pattern but whether they are using the dark red banding as a foraging signal is not clear from these results, suggesting any intraspecific signalling in U. lineatus may be complex. Upeneichthys lineatus was the third most abundant fish that interacted with the 3D models, only surpassed by two species that we observed in large schools, T. novaezelandiae and G. subfasciatus. Higher numbers of conspecifics were observed in proximity to the red and black model fish than the white control fish, but we did not observe any differences between the black and red coloured models. It is possible that U. lineatus can recognise the banded pattern but cannot or do not distinguish between the red and black-striped variants, suggesting that the body pattern of U. lineatus encodes mainly achromatic information. The colour of the banding may or may not be important for U. lineatus depending on the context. It may be that a more universally reliable signal is achromatic which is perceivable under all circumstances, whereas when the observer is viewing the goatfish banding from a distance, or where red light is unavailable, the red pattern is less noticeable. We also observed significantly more goatfish foraging near the models than passing by or showing attraction, irrespective of colour, but it was rare that we observed conspecifics eating alongside the models, with most probing the sediment for food with their barbels. Whether foraging increased around the deployed models was not explored in this study, and any future work should consider the inclusion of a negative control. The static nature of the models, and the fact that they were likely not placed in preferred food patches, may explain why we did not see U. lineatus feed, and possibly why there were no differences in the amount of time conspecifics spent around the models. Without stereo-cameras, we were unable to accurately assess fish size although many conspecifics appeared smaller than our model fish. A previous study found that conspecifics will regularly adopt a white colouration when following focal goatfish (Tosetto, Hart & Williamson, 2023), suggesting that red patterning may also be a dominance signal in U. lineatus. These findings are indicative of social learning within U. lineatus populations and further studies that explore the role of colour in this system are warranted. We rarely observed single-species foraging groups amongst adult goatfish but this may be because adult U. lineatus share resource requirements (Powell, 1989). Rather, adult conspecifics could be using the body position and colour as an indicator of nearby food but recognise the dark red banding as foraging success and avoid competition by finding an alternative patch to feed. Intraspecific signals may be more complex than the colour change alone and more dynamic models that combine appropriate food resources would allow us to extrapolate whether colour change is being used as an intraspecific signal in U. lineatus.

Responses of associated heterospecifics to the model fish varied, indicating that interspecific signalling may be multifaceted and driven at a species level. No difference in the abundance or behaviour of associated heterospecifics in response to the different coloured models was observed but there were some differences in behaviour unrelated to model colour. Pseudocaranx georgianus passed the models significantly more than fed or were attracted, but as a mid-water schooling species that don’t naturally occur on rocky reefs (Folpp et al., 2013) this is not surprising. Atypichthys strigatus were clearly attracted to the models but no feeding was observed. Atypichthys strigatus are zooplanktivores that commonly respond to disturbance events to feed in the water column (Glasby & Kingsford, 1994; Kuiter & Kuiter, 1997). It is likely they were attracted to the head down feeding position of the models, but the absence of disturbance may have meant no food was made available. We also observed two wrasse species, O. lineoata and E. angustipes, displaying attraction to and passing by models more than we observed them feeding. Wrasse are the most opportunistic carnivores in exploiting fishes that foraging through sediment (Aronson & Sanderson, 1987). Like A. strigatus the feeding position, regardless of colour, may provide an initial attraction for the wrasse but the absence of foraging means infauna are not disturbed and subsequently, there are no food resources to exploit. Including sediment disturbance in these models would likely provide insight into the motivation of these follower fishes. However, if the dark red banding of U. lineatus is indeed a signal then we would expect to see a difference in response to the different models regardless of food availability. It is possible that the foraging signal is multifaceted, but it is likely that these species respond to the nose down feeding position of any benthic fish in the anticipation of food.

Assessing the time heterospecifics spent around model fish provides greater insight into associations and potential signals. Both O. lineolata and A. australis are benthic foragers and both spent more time around the white model fish, contrary to our expectations. Due to the opportunistic nature of wrasse (Sazima et al., 2005), it is possible they are attracted to any fish exhibiting a nose down feeding position. A. australis are generalist feeders and will often feed in shallow waters during daytime hightide to avoid predation (Gannon et al., 2015; Taylor, Becker & Lowry, 2018) and possibly do not rely on increased food or vigilance from interspecies group feeding. Furthermore, given coastal fish are often dichromats lacking the visual pigments to see in the longer (red) wavelengths (Marshall, Carleton & Cronin, 2015) and some species of marine fish have excellent contrast sensitivity (Santon, Münch & Michiels, 2019), the white model may have been more obvious to opportunistic fish. We observed no variation in time spent around the models for P. georgianus or A strigatus but as they are both predominately planktivores, the absence of food meant there was little benefit spending time in proximity to the model. Surprisingly, the blacksaddle goatfish P. spilurus did not differ in time spent around different models either. Personal observations suggest there is dietary partitioning as P. spilurus were often seen foraging on rocky reefs rather than bare sand, and regularly seen in schools, thus there may be no extra foraging or vigilance benefits gained from foraging with U. lineatus.

Three species of fish that spent significantly more time around the red and black model fish than the white control fish were G. subfasciatus, S. granulatus and juvenile C. auratus, all of which are benthic carnivores (Edgar, 1997). We observed that G. subfasciatus spent more time interacting with the black-banded model fish, suggesting a response to the banded pattern. While G. subfasciatus and U. lineatus both forage over bare sand it is possible they are not targeting the same prey. Dietary partitioning has been demonstrated between G. subfasciatus and U. tragula (bartailed goatfish) via different feeding strategies. Gerres subfasciatus use their highly protrusible mouth to target surface polychaetes, while goatfish use their barbels to extract prey living deeper below the substrate surface (Linke, Platell & Potter, 2001). Perhaps U. lineatus expose infauna that G. subfasciatus can then exploit from the surface, thereby providing benefits to both fish without competition for resources.

Two fish that spent a significantly higher proportion of time around the red-banded goatfish models were S. granulatus and C. auratus. There is little information available on the visual ecology of S. granulatus but they are habitat specialists, predominately found on seagrass (Harvey et al., 2013). In this study, we regularly observed S. granulatus with the model fish when they had been inadvertently placed on seagrass beds. Given seagrass depth limit is related to light attenuation underwater (Duarte, 1991), it is possible S. granulatus possesses visual pigments sensitive to a broad range of wavelengths present in shallow water, including longer wavelengths, and can perceive the red colouration. The benefits to S. granulatus associating with goatfish are not clear but despite seagrass edges providing greater food availability (Macreadie et al., 2010) prey species will regularly use the interior of seagrass patches for protection from predators (Smith et al., 2011). Interestingly, in Port Phillip Bay on Australia’s south coast the bluestriped goatfish (Upeneichthys vlamingii) and bridled leatherjackets (Acanthaluteres spilomelanurus) are regularly found in deeper seagrass beds (3–6 m) (Smith, Jenkins & Hutchinson, 2012). Perhaps the foraging activity of U. lineatus provides prey to S. granulatus allowing them to remain in the safe interior of the seagrass patch. Juvenile C. auratus are also regularly found foraging with U. lineatus over exposed sand flats adjacent to rocky reef habitats, a distribution thought to balance requirements of food acquisition and predator protection (Ross et al., 2007). It has been reported that juvenile C. auratus have visual pigments with sensitivity in the longer wavelength area of the spectrum (around 570 nm) (Robinson et al., 2017) suggesting the capacity to discriminate the colour red. Perhaps juvenile C. auratus are using the red patterning of the goatfish to indicate suitable food resources whilst also benefiting from increased predator protection. Furthermore, adult snapper also adopt a red body colouration (Booth et al., 2004). Skin pigmentation requires a dietary source of carotenoids (Doolan et al., 2008) which likely come from crustaceans found in the benthos. Given that U. lineatus likely obtain carotenoids from their diet, it may be that C. auratus also follow goatfish as a source of the quality food that they require. It is worth noting that there are some outliers in these data that may have influenced the results. More data are required to strengthen the associations that individual species may have with U. lineatus. There is an exciting opportunity to investigate the drivers of this relationship in more detail and gain insight into some of the important associations and the complexities in signalling taking place in coastal temperate environments.

The increased time that some fish spend around the red-and black-banded models suggests they were responding to the changes in colour and pattern, but without associated bottom disturbance it is difficult to tease out the benefits for associated heterospecifics. Early studies on multi-species foraging associations suggest follower fish are initially anticipatory in their behaviour, following nuclear fish whenever they see them regardless of their behaviour (Strand, 1988). A more recent study (Krajewski, 2009) suggests the initial driver of attraction is bottom disturbance, regardless of the source. In both cases it is reported that fish will only forage with a nuclear fish when there is evidence of them feeding successfully. To date, any examination of group feeding featuring goatfish as the nuclear fish has focused on bottom disturbance as the social cue. Two goatfish species regularly assessed in foraging associations are Pseudupeneus maculatus (spotted goatfish) and Parupeneus barbernius (dash-and-dot goatfish) (Krajewski, 2009; Lukoschek & Mccormick, 2002; Sazima et al., 2006; Sazima et al., 2007). Both are anecdotally reported to change colour, but any changes in colour have not been considered in foraging associations to date. It is possible that bottom disturbance and foraging signals, such as visual cues, are not mutually exclusive and foraging signals reinforce the initial attraction to disturbance. To explore this relationship, (Krajewski, 2009) used models of P. maculatus and disturbed the underlying sediment but found no interaction between the visual appearance of the fish and disturbance. However, changes in goatfish colour were not considered, just the neutral appearance of the fish. Our current study focused on the possible signal of the dark red banding but did not include any bottom disturbance. If the two are not mutually exclusive, then it is essential that any studies combine potential foraging signals with more dynamic models.

The quality of the signal, whether chromatic or achromatic, could be influenced by the intensity of the banding. The rapid colour change of U. lineatus from white to dark red is continuous, shifting from a white to paler red when searching for food and the adoption of the dark red colouration when actively feeding (Tosetto, Hart & Williamson, 2023). It is possible that the intensity or saturation of the dark red bands may indicate the food patch quality. Alternatively, the capacity to turn a deeper red may vary within individuals with the fish that turn darker red indicating better body condition. There are reports showing that some species of fish prefer ‘redder’ individuals. For instance, male cyprinid fish (Puntius titteya) prefer redder males (Fukuda & Karino, 2014), female three-spined sticklebacks (Gasterosteus aculeatus) use the intensity of red colouration in males as an indicator of functional fertility (Pike et al., 2010), and male two-spot gobies (Gobiusculus flavescens), prefer to mate with females with bright yellow-orange bellies (Amundsen & Forsgren, 2001). In situations where ‘red’ is not perceivable, an enhanced saturation of the red banding would increase the contrast between the red and white vertical bands, potentially enhancing the conspicuousness of the achromatic signal. When signalling to potential mates and to facilitate spawning, Paralabrax maculatofasciatus (kelp bass) enhance the contrast of their vertical dark bars (Erisman & Allen, 2005) and in sword tails (Xiphophorus sp.), vertical banding on males intensifies during social interactions (Morris, Mussel & Ryan, 1995). Future studies that seek to test whether the dark red colouration could be indicating the quality of food, or individual body condition are certainly warranted.

A diversity of fish can rapidly change colour and increasingly we are learning that the drivers of this are varied. Camouflage remains one of the most studied drivers of colour change and its main function appears to be for signalling. In some cases, dynamic colour change can be used for multiple reasons such as Nassau groupers (Epinephelus striatus) that use colour change for camouflage (Watson, Siemann & Hanlon, 2014) as well as social signalling (Archer et al., 2015). In African dwarf chameleons (Bradypodion spp.) colour change is regularly used in camouflage but it evolved as a conspicuous signal in male contest (Stuart-Fox & Moussalli, 2008). There is anecdotal evidence to suggest U. lineatus adopt a red colour at night, possibly for camouflage while resting. It is possible this colour change is multifunctional, which may have evolved for camouflage and has been co-opted for use in social signalling. Alternatively, the selection for conspicuous social signals may have driven the evolution of colour change in these goatfish. There are anecdotal reports of several other goatfish species having the capacity to rapidly change colour but to date this has rarely been investigated. There is only one other study that we are aware of that quantifies colour change in goatfish. In a mutualistic signalling event, M. martinicus (yellow goatfish) and P. maculatus (spotted goatfish) will shift body colouration, appearing darker to cleaner shrimp and indicating their willingness to be cleaned (Caves, Green & Johnsen, 2018). These findings demonstrate that different species of goatfish use rapid colour change for signalling but in different contexts. Such signals and sensory systems of receivers may have evolved in response to changes in local environments (Endler, 1992).

Conclusions

A total of 58 different fish species interacted with the 3D models. U. lineatus was observed in higher abundance around the coloured model fish than the white control model fish suggesting U. lineatus could recognise the banded pattern but did not distinguish between the red- and black-coloured variants, suggesting that the body pattern of U. lineatus encodes mainly achromatic information. The lack of goatfish foraging near the models could be due to the static nature of the model fish. Alternatively, it could suggest competition for resources between conspecifics. Heterospecifics spent longer time periods around the red and black model fish than the white control fish. Because the model fish were static and did not increase the food availability in the benthos, it was difficult to obtain conclusive results as to whether follower fish benefited from this signal by obtaining access to food resources or were benefiting purely from increased protection. Including different coloured models that incorporate movement and bottom disturbance would provide greater insights into drivers of colour change and subsequent benefits to associated fishes.

Understanding which species in temperate systems are responding to the foraging signal of U. lineatus, as well as understanding the benefits received, will provide greater understanding into this signalling system. It is likely that heterospecific fish obtain additional food resources by foraging with goatfish but this requires further in-situ research to tease this apart. Further, coupling such research with an assessment about the visual systems of different signal receivers will provide greater insight into the complexities and coevolution of signalling systems. Given that there is evidence of other goatfish species using colour change as a signal means there are exciting opportunities to explore the evolution of rapid colour change using the goatfishes (Mullidae) as a model family.

Supplemental Information

Supplemental Information 1 Supplemental Figure and Tables

Click here for additional data file.

Video S1 3D Model Goatfish

Animated version of the wavefront.obj file that was generated for the 3D printed goatfish model. Model was created using Cinema4D. 3D Model Credit: Mike Tosetto @ Never Sit Still

Click here for additional data file.

Thanks to Michael Tosetto from Never Sit Still for designing and modelling the 3D model fish, and David Connelly for extensive experience in designing and building the RUVs. Thanks to members of the Marine Ecology Group, the Fish Lab, Macquarie University technical support staff, especially Nick Harris, for extensive field assistance, Claire Populus for extensive video analysis, Andrew Allen for his guidance with statistical analysis and Patrick Burke for his help with GIS. We acknowledge the Gadigal and Dharawal people of the Eora Nation, the Dharawal-Dhurga people of the Tharawal Nation, the Thaua people of the Yuin Nation and Wallumattagal people of the Dharug Nation who are the Traditional Custodians of the land on which we worked. We recognise their continuing connection to land, water and community. We pay respect to Elders past, present and emerging.

Additional Information and Declarations

Competing Interests

Author Contributions

Animal Ethics

Field Study Permissions

Data Availability

The authors declare there are no competing interests.

Louise Tosetto conceived and designed the experiments, performed the experiments, analyzed the data, prepared figures and/or tables, authored or reviewed drafts of the article, and approved the final draft.

Nathan S. Hart conceived and designed the experiments, authored or reviewed drafts of the article, and approved the final draft.

Jane E. Williamson conceived and designed the experiments, authored or reviewed drafts of the article, and approved the final draft.

The following information was supplied relating to ethical approvals (i.e., approving body and any reference numbers):

Macquarie University Animal Ethics Committee provided full approval for this research (ARA 2016/020).

The following information was supplied relating to field study approvals (i.e., approving body and any reference numbers):

Field experiments were approved by the Macquarie University Technical Staff

The following information was supplied regarding data availability:

The datasets are available at FigShare: Tosetto, Louise; Hart, Nathan; Williamson, Jane (2023). A fish can change its stripes: investigating the role of body colour and pattern in the bluelined goatfish. Macquarie University. Dataset. https://doi.org/10.25949/23648031.v1.

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
