# Peer review of "A fish can change its stripes: investigating the role of body colour and pattern in the bluelined goatfish"

_PeerJ, doi:10.7717/peerj.16645_

## Round 0.1 · original submission · Minor Revisions

In line with the comments and suggestions of the reviewers, I agree that the manuscript is written in clear language and a solid framework. I welcome the authors to prepare a point-to-point rebuttal and look forward to reading the revised manuscript.

Reviewer 1 ·

Basic reporting

The manuscript of Tosetto et al. investigates the potential role of color change in bluelined goatfish using 3D model goatfish with different color morphs. The overall experimental design is interesting and answers the hypotheses of the authors: does color play a role in food signaling to other conspecifics or associated heterospecifics. Language used is clear, unambiguous, and easy to follow. I especially liked the introduction section, as the authors provided sufficient background information regarding their research topic, and the flow is interesting.

Experimental design

The overall experimental design is extensive and well-structured to answer their research question. The meticulous procedures in the preparation of the 3D model fish are commendable. However, it would be nice if the authors would also include an example of the 3D models with black and white bands, and white models (only the red and white models are shown).
However, I would suggest the authors to clearly specify (1) the number of recordings made, (2) durations and intervals of the recordings, (3) and time of the recordings.
Since the experiments were conducted in a 13-month period (line 199), will seasons be a potential factor affecting the observed species that were interacting with the 3D models? Also, different conspecific and heterospecific species might have different feeding period, the authors should make clear whether the experiments were conducted at the exact same time for each recording?
Also, kindly indicate how different heterospecific species were identified (line 229-233). Any taxonomic keys were used?

Validity of the findings

The discussion is in-depth and justifies the results. I have, however, one query - since heterospecifics showed attraction to the models (line 341-344) regardless of model color, would it be wise if the authors add another control, where observations of the interactions between real bluelined goatfish and heterospecifics are observed?

Additional comments

Line 67 & 77: are these two in-press articles the same? Because one is marked with year (2023) and one is not.
Line 126: do you mean 'patterning'?
Add scale to Figure 1 & 2.

Reviewer 2 ·

Basic reporting

Generally, this manuscript has reported an interesting work. It was nicely constructed. The introduction was well written, the objectives were clear, and the methodology, results, and discussion were also done nicely. However, there are several questions/ recommendations to enhance this work further.

Experimental design

1. When were the recordings conducted? Were they conducted in the morning, noon, or afternoon? Also, it is best if the water quality condition, especially turbidity level or visibility and underwater light intensity can be provided in the text as turbidity level may affect light penetration and underwater light condition.

2. Any specific reason why only one sampling site at Kioloa and Eden?

Validity of the findings

1. In this study, the parametric statistical test has been used. In this case, it is necessary to explain the outcome of the data homogeneity and normality tests.

2. Any differences in the behavioral data collected from different locations, such as between Jervis Bay and Sydney?

Additional comments

Generally, this manuscript has reported an interesting work. It was nicely constructed. The introduction was well written, the objectives were clear, and the methodology, results, and discussion were also done nicely. However, there are several questions/ recommendations to enhance this work further.

Reviewer 3 ·

Basic reporting

The paper is well written and it is easy to follow what were the context, the aims and the methods to assess the hypotheses.
- Figure 5&6: Good representation of the raw data, but very difficult to assess which interactions or differences might be meaningful. Could you add indicators or model-based means and 95% confidence intervals? You could add sample sizes per group in the caption.

Experimental design

The experimental design is easily understandable from the descriptions and appropriate. There are only a few things to clarify:
- Ln 210: Your locations were in less than 10 m, but how different were they in terms of depth? A few meters can already make a difference, also in this depth (e.g. I would expect a strong difference in how red the models appear comparing 3 and 10 m). In this regard, is depth similar at least per site? I expect you have strong site-specific effects. Potentially also because models were located quite close to each other. Are 5 m distance enough? Many fish will have seen all models at one site. Therefore, you might also have an order effect, with fishes that already explored a model not being so interested in the others anymore. There might be an effect that fish spotted and interacted with the red or black models first because they were more conspicuous and were then habituated, not interacting with the white model so much anymore.
- Ln 286: what’s the response in the original model? Model colour? It’s not really clear from the phrasing.
- Ln 291: what exactly is abundance? You do not specify this in the paragraph starting in line 240. I assume it’s the total number of interactions summing up all three behaviours? It would be good if you can clarify this at some point. What is the model distribution family? Gaussian?
- Table 1: It would be great to have the behaviours in here as well. Like this, the reader cannot assess the sample size per behaviour, or at least I did not find it somewhere else.

Validity of the findings

The data seems robust, but discussion and interpretations need a more rigorous and careful explanation:
- Ln 70: you say that fish with the red morph are seemingly more conspicuous. This seems like a simple explanation for the result that hetero- and conspecifics are more abundant with the black and red models without the morphs necessarily having a signalling function.
- Ln 378: they also were foraging more than the other species, right? But this could be caused by general differences in their foraging time. A negative control without model is missing, which would be interesting to see whether foraging is actually increased around a model.
- Ln 407-409: It is unclear how this can be concluded. What findings exactly are meant? Are you referring to the fish size mentioned in the previous sentence? You cite a paper in press, maybe indicate what exactly was found here, as the reader cannot access the paper. Or are you referring to the finding that there is a higher abundance at the black and red models? How does this indicate social learning? A more detailed explanation would be great.
- In the same context, Ln 381; 407-409: see my comment above about the camouflage – is it possible that there was higher abundance of conspecifics because the black and red models were simply more conspicuous?
- Ln 460-470: you explain well why it would be beneficial for certain fishes to feed with the focal species. But how does this link to the differences in time spent with the different models? Why would certain species have other preferences than others? It seems like some species (like the wrasses) prefer the white models and others the dark models. Why such different preferences? What might be the link to conspicuousness?
- Ln 495: what is the link to their own body colouration and high quality food from the goatfish? Is the only source of high quality food needed to achieve the red colouration the food source of the goatfish? I can’t follow the argument.
- Ln 554: What would be the alternative? Maybe colour change in the end did evolve only for camouflage. Is it possible that during feeding, they just fail to stay white for some reason? On a sandy patch, I would think that their white morph makes a better background match. So maybe their dark banded pattern is the colour morph during resting (as it is while they sleep) and only the active dispersion of pigments in their chromatophores allows them to appear white. During feeding, maybe they don't have the physiological capacity to stay white and becoming darker is just a side effect. Of course, this is also just hypothetical, but it would be good if you can rule out such explanations, even if it is just by providing examples from other species etc., or at least discuss them briefly.
- Ln 93-95: You mention the idea that the colour change indicates food patch quality here in the introduction. I would remove it and only add it in the discussion because here, it appears like a hypothesis that you would like to test, which you don’t do in this paper. In general, I think it would be extremely interesting to have an experiment like this, and it would also much more support the hypothesis that the colour change is indeed a social signal than the evidence that you present here. I think the discussion would greatly benefit from restructuring, discussing why fish should benefit from foraging with the focal species; discussing what indicators you now actually have that this might be true (more interactions with red/black models in some cases), but what would be alternative explanations to this observation (!); and then what would be the next steps.
- Ln 399 and paragraph starting ln 472: I think you definitely found out that the colour of the fish is not so important, at least for conspecifics – this is super interesting and yet you rarely discuss it! What’s the benefit? You mention this in the introduction but I think it is the clearest result and therefore also should be discussed again.

Additional comments

- Figure S1: The caption is missing; what spectra are shown here? Is A) the real goatfish? And in C), where is the black spectrum? Is it the line that looks like the x-axis? Maybe it would be easier if the line was dashed or had a different colour.
- Ln 120: Blue light is not only lost in depth but also in the horizontal, meaning the chromatic signal is probably also lost in distance, another reason why it would likely be an achromatic signal.
- Ln 229: Typo in lineolatus
- Ln 255 Typo any = and?
- L 331: “total interactions” = abundance? See comment above, this is unclear. It is difficult to understand the difference to the first phrase in the paragraph (ln 326), this means that there was no effect of the specific behaviours being more or less displayed depending on model type, but when looking at abundance, there is an effect? Maybe the whole paragraph would benefit from rephrasing. The next paragraph (ln 337) is much easier to understand.
- You mention the “coloured” models several times (e.g. ln 382). Sometimes it is not immediately clear what you mean, the fish with the stripes only or all three? Maybe you find a better phrasing (e.g. just “the different models”)
- Ln 437: you would expect to see a difference in response to the different colours. Maybe rephrase to make the sentence easier to follow
- Ln 445: This is inconclusive, if they are attracted to any model, there would be no difference and not a preference to stay longer with the white models, no?
- Ln 451: Why more attractive? The fact that many fishes are dichromats only means that the red and black model both will appear darker than the white one. And why should fish prefer white models when they can perceive the contrast of the stripes better? Please try to clarify your argumentation.
- Ln 497: There seems to be a word missing: “that individual … ? may have”; individual species?
- Ln 533: With the possible preference you are referring to mate choice, right? Is it reasonable that they signal for mate choice during feeding? Can you give examples like this from other species or is this known from goatfish? Usually you would expect such displays during social interactions, as you also indicate at the end of the paragraph.

---

## Round 0.2 · accepted · Accept

I agree with the reviewers that this manuscript is now ready for publication.

Reviewer 2 ·

Basic reporting

The authors have revised and improved the manuscript according to the reviewers' comments and suggestions. The manuscript is now recommended to be accepted for publication.

Experimental design

no comment

Validity of the findings

No comment

Reviewer 3 ·

Basic reporting

no comment

Experimental design

no comment

Validity of the findings

no comment